# A Systematic Evaluation of Out-of-Distribution Generalization in Crop Yield Prediction

**Aditya Chakravarty**
**Independent Research**
`chakravarty.aditya28@gmail.com`

Reviewed on OpenReview: `https://openreview.net/forum?id=to4sVjsxsO`

## Abstract

Accurate crop yield forecasting under shifting climatic conditions is essential for food security and agricultural resilience. While recent deep learning models achieve strong performance in in-domain settings, their ability to generalize across space and time—critical for real-world deployment—remains poorly understood. In this work, we present the first systematic evaluation of temporally-aware crop yield prediction models under spatio-temporal out-of-distribution (OOD) conditions, using corn and soybean data across more than 1,200 U.S. counties. We benchmark two representative architectures, GNN-RNN and MMST-ViT, using rigorous evaluation strategies including year-ahead forecasting, leave-one-region-out validation, and stratified OOD scenarios of varying difficulty based on USDA Farm Resource Regions. Our comprehensive analysis reveals significant performance gaps across agro-ecological zones, with some models showing negative $R^2$ values under distribution shift. We uncover asymmetric transferability patterns and identify the Prairie Gateway region as consistently challenging for generalization. These findings challenge prior generalizability claims and provide practical insights for deploying agricultural AI systems under climate variability.

## 1 Introduction and Related Work

Crop yield prediction is a critical task in agricultural planning, food security, and economic forecasting, particularly in the face of climate change (Houghton et al., 1990). Traditional process-based models, while useful, often suffer from high inaccuracies due to strong assumptions about management practices and high computational costs (Leng and Hall, 2020). Studies have demonstrated the growing impact of climate change on agricultural productivity (Ortiz-Bobea et al., 2018), (Zhao et al., 2017). These challenges have driven the adoption of machine learning (ML) and deep learning (DL) approaches, which leverage large-scale environmental and satellite data for more accurate crop yield predictions. Recent IPCC reports underscore the growing and detrimental effects of climate change on global agricultural yields, further emphasizing the need for adaptive strategies and accurate predictive models (IPCC, 2021). Recent deep learning models improve crop yield

prediction by capturing spatio-temporal dependencies, but true spatial out-of-distribution (OOD) generalization remains unexplored. Bridging this gap is critical for forecasting yields in regions lacking historical data or facing climate-driven shifts. This study addresses this critical gap through systematic evaluation of model robustness under realistic spatio-temporal distribution shifts.

**DL models in yield prediction** Deep learning (DL) models for crop yield prediction leverage two complementary types of data sources: remote sensing inputs (satellite imagery, UAV data, vegetation indices) and meteorological/environmental inputs (weather variables, soil parameters). While earlier works often relied on one or the other, state-of-the-art models increasingly fuse both modalities, as they capture different aspects of crop growth dynamics. This distinction is useful for understanding model design, though in practice the two input types are complementary rather than mutually exclusive.

Early neural network models for crop yield prediction demonstrated promising results, outperforming conventional regression techniques (Drummond et al., 2003), (Liu et al., 2001). Among the meteorological and environmental data-based approaches, a key advancement was the CNN-RNN framework, which integrates multi-year meteorological and environmental data to improve yield forecasts (Khaki et al., 2019), (Khaki and Wang, 2021). This method established the importance of historical weather data, demonstrating that using multi-year sequences of climate variables significantly enhances prediction accuracy.

Building upon CNN-RNN architectures, newer methods incorporate graph neural networks (GNNs) to model geographical dependencies. The GNN-RNN model extends CNN-RNN by incorporating spatial relationships among counties, enabling the model to leverage information from neighboring regions to refine yield predictions (Fan et al., 2022) using long-term meteorological data. This method has shown improvements over CNN-RNN models in various evaluations, demonstrating the benefits of integrating spatial context into deep learning frameworks.

On the other hand, remote sensing-based approaches utilize satellite and UAV-derived data to estimate crop yields (Tseng et al., 2021). These methods analyze features extracted from spectral indices, vegetation health, and high-resolution imagery to make yield predictions. While effective, many of these models overlook the direct impact of meteorological variations on crop growth (Sainte Fare Garnot and Landrieu, 2021).

More recently, hybrid approaches have emerged that integrate both remote sensing data and meteorological variables for enhanced crop yield predictions. As climate change increasingly affects global agriculture, models that explicitly incorporate long-term climate trends have gained attention (Ortiz-Bobea et al., 2018), (Zhao et al., 2017). Unlike earlier DL models that focused primarily on short-term weather patterns, transformer-based architectures explicitly model long-term climate shifts, enabling stakeholders to assess their effects on crop yields (Lin et al., 2023). By leveraging attention mechanisms, these models both short-term weather fluctuations and long-term climate trends, offering an alternative approach to GNN-based methods.

## 2 Motivation and Contribution

Despite growing interest in deep learning for crop yield prediction, systematic evaluation of model robustness under realistic distribution shifts remains limited. Existing studies typically evaluate models within the same geographic regions used for training, providing an incomplete picture of

real-world deployment scenarios where models must generalize to new climatic conditions, unseen regions, or future time periods. This evaluation gap is particularly concerning for agricultural applications, where climate change is continuously shifting the underlying data distribution and reliable forecasts are needed for regions with limited historical data.

We conduct a systematic comparative analysis across both temporal-only and spatio-temporal OOD settings, using data from over 1,200 counties in the top 20 U.S. corn and soy producing states. Counties are grouped into scientifically grounded clusters based on geographic and climatic characteristics, enabling rigorous testing of generalization. Furthermore, we design deployment scenarios categorized by cluster similarity to assess model robustness across OOD shifts of varying severity, with direct implications for climate-resilient agriculture and food security. By using publicly available datasets and scientifically robust clustering techniques, this work not only enhances the understanding of model generalizability in practical agricultural forecasting scenarios but also informs strategic model deployment decisions under climate variability. Our findings provide insights into the relative strengths and limitations of MMST-ViT and GNN-RNN and offer guidance for selecting optimal predictive approaches under different spatial-temporal distribution shifts.

## 3 Dataset and features

We used the CropNet dataset (see Lin et al., 2024, [1]), which is a large-scale, publicly available, multi-modal dataset specifically designed for climate change-aware crop yield predictions across the contiguous United States from 2017 to 2022. Table 8 summarizes the CropNet dataset components.

## 4 Experiment Design

This study investigates two forms of out-of-distribution (OOD) generalization: temporal OOD and spatio-temporal OOD. Temporal OOD refers to year-ahead prediction, where models are trained on data from 2017–2021 and tested on 2022 in the same region. Spatio-temporal OOD corresponds to a more challenging : the test set is not only for 2022 but also from a geographic region entirely excluded during training. This dual-axis domain shift—across both time and space—forms a stringent test of model generalizability.

Each experiment in Table 1 is designed to isolate a specific aspect of OOD robustness: UMAP validates the clustering basis; reproduction establishes baselines; LCO-CV measures spatial generalization; ablations quantify each modality's contribution; the pairwise matrix reveals asymmetric transfer; and deployment scenarios simulate real-world conditions of varying difficulty.

**Data Preparation**

**USDA Farm Resource Regions for Robust Regional Classification and OOD Evaluation**
The counties in our study were classified into seven clusters based on the USDA Farm Resource Regions (Heimlich, 2000), a widely recognized and scientifically validated framework for agricultural classification. These regions are delineated using climate, soil properties, and topographic characteristics, ensuring that counties within the same region share similar agro-environmental conditions. Unlike arbitrary spatial partitioning, this classification method captures the biophysical

---

[1] https://huggingface.co/datasets/CropNet/CropNet

Table 1: Experimental design and guiding questions for evaluating OOD generalization. Each experiment is necessary to isolate a specific aspect of OOD robustness; together they form a comprehensive benchmarking protocol.

| Experiment | Description | Question |
|---|---|---|
| UMAP embedding | Cluster counties using UMAP on weather variables | Spatial structure; Are USDA FRRR valid basis for OOD analysis? |
| Reproduce prior models | GNN-RNN and MMST-ViT runs using original configs and datasets | Do original performance claims hold under re-evaluation? |
| Leave-one-cluster-out CV | Tune models via LCO-CV and assess generalizability across clusters | Which regions fit/unfit for spatio-temporal generalization ? |
| Ablation: No message passing | Disable message passing in the GNN-RNN model | How important is spatial message passing in GNN-RNN? |
| Ablation: No Sentinel-2 input | Remove Sentinel input from MMST-ViT, only use HRRR data | What role does remote sensing play in model accuracy? |
| Region pairwise matrix | Evaluate models on all train-test region pairs | How does performance vary across region combinations? |
| Real-world deployment scenarios | Define OOD cases using LCO-CV results, geography and UMAP | How well do models generalize in realistic deployment settings? |

and agronomic factors that influence crop production, making it a robust basis for regional analysis (Spangler et al., 2020). Figure 1 illustrates the spatial distribution of study areas.

By leveraging the USDA Farm Resource Regions, we ensure a scientifically grounded approach to OOD evaluation, essential for building robust agricultural AI models that remain reliable under future climate uncertainties. For all experiments, data from 2017 to 2021 is used for training, while models are evaluated using 2022 data. All experiments were conducted on a system with NVIDIA RTX 4090 GPU (24 GB VRAM) and 64 GB RAM.

**Architecture Tuning and Year-Ahead Evaluation**  Model sensitivity analysis was conducted during the leave-one-cluster-out cross-validation (LCO-CV) phase to select optimal hyperparameters and identify regions with strong generalization potential. For GNN-RNN, we varied the number of layers (2, 3, 4), aggregation type (mean, pool, gcn), and dropout rate (0, 0.5, 0.7). For MMST-ViT, we explored embedding dimensions (128, 512) and dropout rates (0, 0.3, 0.5). The best model configurations from LCO-CV were carried forward into year-ahead evaluations.

In this setting, models were trained on data from 2017–2021 and tested on 2022, with each train-test region pair evaluated independently. For MMST-ViT, this included re-running both contrastive pretraining and fine-tuning. Diagonal entries in the resulting performance matrices represent within-region year-ahead forecasting; off-diagonal entries quantify spatio-temporal transfer. To simulate realistic deployment, train regions were grouped into larger sets than the test regions (3-to-1 splits), with test candidates selected based on generalization insights from LCO-CV. The evaluation spans seven USDA Farm Resource Regions (EU, HL, MSP, NGP, NC, PG, SS), with RMSE and standard deviation reported over three random seeds for both corn and soybean.

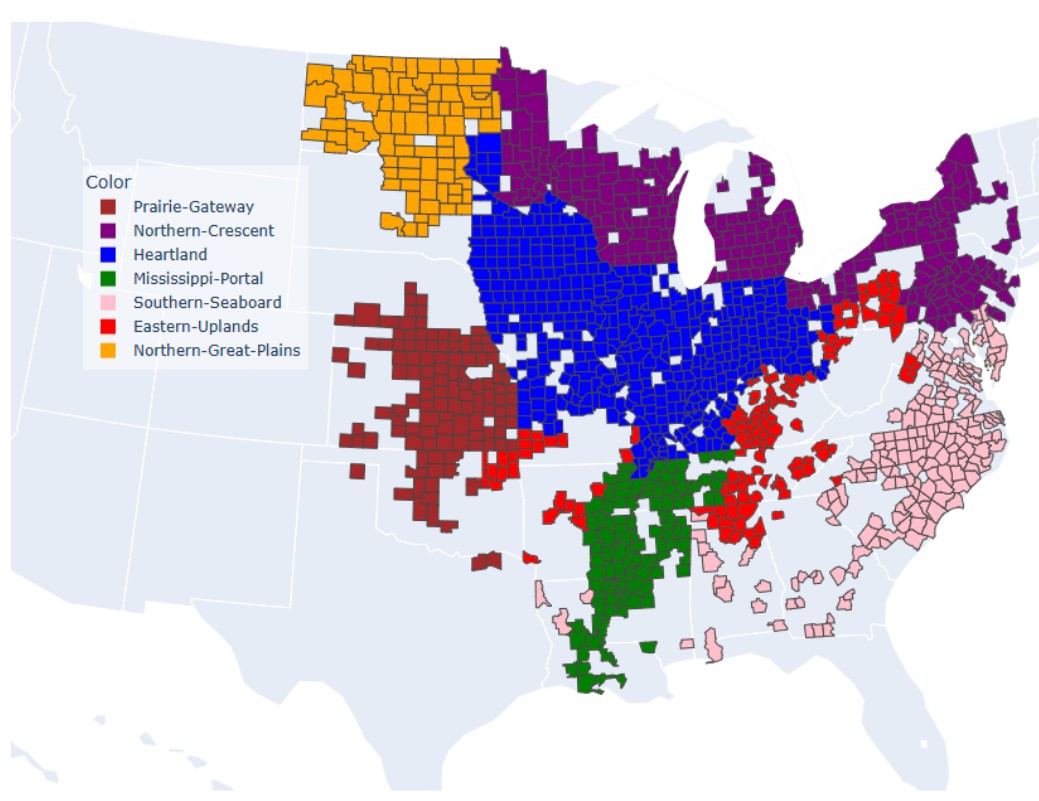

Figure 1: Spatial Distribution of Study Areas and USDA Farm Resource Regions

## 5 Results

### 5.1 Reproduction of Published GNN-RNN and MMST-ViT Results

To validate our implementation pipeline, we first reproduced the GNN-RNN model (Fan et al., 2022) using the same architecture and hyperparameters as reported in the original paper. As only data from Illinois (IL) and Iowa (IA) were publicly available, our experiments were limited to these two states. We trained the model on data from 1981–2017 and evaluated on the 2018 season.As shown in Table 2, our reproduced RMSE/Std-dev values differ by only 0.04 for soybean and 0.05 for corn from those reported. The $R^2$ and correlation values are also within 0.01–0.09 of the published results.

Table 2: Comparison of GNN-RNN and MMST-ViT performance between our reproduced results and those reported in the original publications.

| Model (Dataset) | Crop | Present study | | | Reported in original paper | | |
|---|---|---|---|---|---|---|---|
| | | RMSE/Std-dev | $R^2$ | Corr | RMSE/Std-dev | $R^2$ | Corr |
| GNN-RNN | Soybean | 0.43 | 0.72 | 0.83 | 0.47 | 0.73 | 0.86 |
| (only IL + IA) | Corn | 0.41 | 0.58 | 0.73 | 0.46 | 0.67 | 0.82 |
| | | RMSE | $R^2$ | Corr | RMSE | $R^2$ | Corr |
| MMST-ViT | Soybean | 6.98 | 0.81 | 0.87 | 5.00 | 0.89 | 0.89 |
| (IL + IA + MS + LA) | Corn | 17.61 | 0.78 | 0.84 | 10.00 | 0.89 | 0.89 |

We also reproduced the MMST-ViT model (Lin et al., 2023) using the same dataset (IL, IA, MS, and LA), input modalities (Sentinel-2 and HRRR), and training workflow, including both the contrastive pretraining and fine-tuning stages. All implementation details and data preprocessing steps were kept identical to those outlined in the original paper. Nevertheless, our reproduced RMSE values were higher by 1.98 for soybean and 7.61 for corn (in bu/acre), with $R^2$ values lower by up to 0.11. Correlation differences remained within 0.05. While some variation is expected due to factors such as random initialization or nondeterministic GPU operations, the magnitude of the discrepancy raises the possibility that not all aspects of the original experimental setup were fully documented or reproducible from the available information. All experiments in the present study were repeated three times with different random seeds, and we report the mean performance across runs. In contrast, our GNN-RNN model—trained without soil and management variables—achieved performance nearly identical to the original paper that included these features, indicating that such inputs may contribute little additional value in this specific setting.

### 5.2 Cross validation, Ablation outcomes, and real-world train-test scenarios

Overall, performance in both settings was substantially weaker than in-distribution evaluations. Year-ahead predictions (temporal OOD) showed moderate degradation in $R^2$ and correlation, but performance remained usable in several regions. In contrast, the spatio-temporal LORO setting resulted in significantly worse outcomes. Across many configurations, models exhibited negative $R^2$ and correlation values, indicating that predictions were not only inaccurate but in some cases inversely related to ground truth.

Eastern Uplands (EU) was the most consistently predictable region across both crops and OOD settings. Heartland (HL) and Northern Great Plains (NGP) also stood out, with multiple GNN-RNN

and MMST-ViT configurations achieving positive $R^2$ values. For soybean, HL achieved $R^2 > 0$ in settings such as `agg_type=gcn`, `n_layers=4`, and MMST-ViT with `e=512, drop=0`. NGP exceeded $R^2 = 0.25$ in several GNN configurations. For corn, NGP again achieved the highest out-of-region $R^2$ ($\approx 0.45$) under MMST-ViT with `e=128, drop=0.5`. Model design choices had clear effects on generalization. For GNN-RNN, higher dropout rates were consistently associated with weaker OOD performance. In particular, for soybean in NGP, $R^2$ dropped from approximately 0.25 (no dropout) to below –0.4 when `dropout=0.7` was used. Increasing the depth to `n_layers=4` was sometimes beneficial, especially in HL and NGP, but did not yield consistent improvements across all regions.

For MMST-ViT, configurations with smaller embedding size (`e=128`) and no dropout (`drop=0`) generally led to the best results. Larger embedding sizes and stronger regularization often resulted in poorer generalization, with several cases showing negative $R^2$ and correlation values—particularly in Prairie Gateway and Southern Seaboard. Given these observations, we selected the following configurations for downstream use and OOD-focused evaluations: for GNN-RNN, `n_layers=4`, `dropout=0`, and `agg_type=mean` or `pool`; for MMST-ViT, `e=128, drop=0`. Table 3 summarizes the training and testing times associated with two model architectures. There is tremendous gap between the two owing to the model design.

Table 3: Training time comparison of MMST-ViT and GNN-RNN on a single RTX 4090 GPU. GNN-RNN achieves a $\sim 135\times$ speedup over MMST-ViT.

| Model | Pretraining Time | Fine-tuning Time | Total Training Time |
|---|---|---|---|
| MMST-ViT | 23 hours | 8.5 hours | 31.5 hours |
| GNN-RNN | | – | 14 minutes |

**Sentinel-2 Data Ablation in MMST-ViT**   Replacing Sentinel-2 inputs with dummy values led to consistent performance drops across crops and regions. This indicated that satellite imagery contributes improves MMST-ViT performance even when weather data is available.

**Message Passing Ablation in GNN-RNN**   Removing spatial message passing-effectively reducing the model to a CNN-RNN baseline without graph-based context led to a noticeable decline in generalization performance. This underscores the importance of graph connectivity in leveraging local context for spatial generalization.

Table 4 summarizes the final training and testing clusters associated with each hypothesized OOD difficulty level. The easy, medium, and hard splits are defined heuristically, based on a qualitative assessment of leave-cluster-out (LCO) cross-validation results, UMAP embeddings, and geographical proximity.

### 5.3   Cross-region transferability and pairwise RMSE patterns

Tables 5 and 6 report the full $7 \times 7$ pairwise RMSE matrices for GNN-RNN and MMST-ViT respectively, across both soybean and corn. These matrices reveal asymmetric transferability patterns that are directly relevant to OOD understanding: the off-diagonal entries quantify how much performance degrades when training and test regions differ. For soybean with GNN-RNN, diagonal values (2022 year-ahead RMSE) are lowest for HL (6.15), MSP (6.09), and NGP (7.11),

Table 4: Real-world scenarios and corresponding USDA Farm Resource Region splits.

| Scenario | Train Region | Test Region |
|---|---|---|
| Case 1 (Easy) | Prairie-Gateway + Heartland + Mississippi-Portal | Eastern-Uplands |
| Case 2 (Medium) | Northern-Crescent + Prairie-Gateway + Northern-Great-Plains | Heartland |
| Case 3 (Hard) | Prairie-Gateway + Southern-Seaboard + Mississippi-Portal | Northern-Great-Plains |

indicating relatively stable temporal generalization. However, the off-diagonal entries reveal strong asymmetries. Models trained on HL generalize reasonably well to MSP (8.88) and NGP (7.73), but not to PG (19.30) or SS (11.50). Similarly, NGP→HL (8.04) and NGP→MSP (9.70) are relatively low, suggesting latent affinity among these midwestern regions. In contrast, training on PG consistently results in high RMSE values across all test regions (e.g., PG→EU: 14.71, PG→MSP: 15.54, PG→SS: 13.03), indicating poor generalization. Notably, SS shows low within-region error (7.96), but generalizes poorly to others. In corn with GNN-RNN, HL again shows strong within-region performance (22.67), with moderately good transfer to MSP (32.44) and NGP (32.20). Training on NC leads to particularly low RMSE in EU (22.29), suggesting that NC-trained models may capture transferable features for EU. In contrast, PG again acts as an unreliable source region, with high RMSE when predicting EU (48.36), HL (33.31), and SS (45.18). PG also exhibits the worst transferability in the soybean setup, indicating structural dissimilarity.

Table 5: RMSE for Soybean (left) and Corn (right) using GNN-RNN model; diagonal entries (bold) represent year-ahead predictions for 2022. Color scale is shown below the table.

| Train\Test | EU | HL | MSP | NGP | NC | PG | SS | Train\Test | EU | HL | MSP | NGP | NC | PG | SS |
|---|---|---|---|---|---|---|---|---|---|---|---|---|---|---|---|
| | | | | | Soybean | | | | | | | | Corn | | |
| EU | **5.46** | 7.24 | 11.55 | 9.01 | 9.63 | 21.42 | 9.70 | EU | **26.69** | 38.92 | 43.55 | 32.86 | 22.29 | 97.42 | 41.77 |
| HL | 8.39 | **6.15** | 8.88 | 7.73 | 8.57 | 19.30 | 11.50 | HL | 33.05 | **22.67** | 32.44 | 32.20 | 33.66 | 56.21 | 49.68 |
| MSP | 17.22 | 11.17 | **6.09** | 9.79 | 9.72 | 26.22 | 10.59 | MSP | 40.98 | **23.48** | **31.03** | 30.87 | 37.03 | 53.04 | 39.05 |
| NGP | 8.94 | 8.04 | 9.70 | **7.11** | 11.88 | 23.37 | 12.48 | NGP | 33.81 | **25.83** | 34.33 | **21.65** | 57.52 | 51.49 | 44.54 |
| NC | 12.25 | 12.39 | 12.67 | 10.29 | **7.25** | 28.23 | 12.51 | NC | 40.13 | 41.67 | 30.56 | 34.64 | **24.42** | 92.17 | 51.94 |
| PG | 14.71 | 11.43 | 15.54 | 14.62 | 13.59 | **11.71** | 13.03 | PG | 48.36 | 33.31 | 42.21 | 41.59 | 52.47 | **42.94** | 45.18 |
| SS | 13.55 | 10.69 | 12.17 | 12.86 | 9.77 | 11.20 | **7.96** | SS | 42.13 | 29.91 | 39.46 | 41.47 | 32.35 | 53.41 | **25.09** |

**Color scale** — Soybean RMSE (bu/acre):  < 8   8–11   11–14   ≥14    Corn RMSE (bu/acre):

< 28   28–40   40–55   ≥55

Turning to MMST-ViT, the year-ahead results are less promising overall. For soybean, HL, MSP, and SS show relatively low diagonal RMSE (10.18, 8.71, and 7.15, respectively), but cross-region RMSEs are considerably higher than those observed with GNN-RNN. For example, HL→NGP is 25.01, MSP→PG is 23.15, and NC→SS is 14.26. Soybean predictions from PG are unreliable across the board, consistent with GNN-RNN findings. Corn performance with MMST-ViT is similarly degraded. Only a few region pairs achieve RMSE below 30, e.g., EU→NC (29.14) and SS→EU (30.22). Diagonal entries for HL (33.37), NC (37.93), and SS (35.48) are modestly acceptable, but generalization across regions is again poor. PG remains the least reliable source region with

Table 6: RMSE for Soybean (left) and Corn (right) using MMST-ViT model; diagonal entries are bold represent year-ahead predictions for 2022 and colors indicate performance. Color scale same as Table 5.

| Train\Test | EU | HL | MSP | NC | NGP | PG | SS | Train\Test | EU | HL | MSP | NC | NGP | PG | SS |
|---|---|---|---|---|---|---|---|---|---|---|---|---|---|---|---|
| | | | | | Soybean | | | | | | | Corn | | | |
| EU | 8.93 | 11.16 | 9.69 | 9.38 | 14.51 | 22.15 | 15.25 | EU | **26.06** | 38.83 | 35.82 | 29.14 | 34.29 | 59.27 | 45.90 |
| HL | 11.93 | **10.18** | 11.89 | 14.34 | 25.01 | 28.22 | 20.58 | HL | 43.70 | **33.37** | 52.20 | 52.20 | 52.38 | 89.87 | 69.84 |
| MSP | 9.62 | 11.91 | **8.71** | 9.72 | 16.87 | 23.15 | 10.78 | MSP | 35.21 | 39.39 | **41.32** | 35.96 | 49.69 | 69.85 | 57.16 |
| NC | 11.34 | 10.45 | 10.13 | **14.49** | 19.58 | 24.76 | 14.26 | NC | 36.32 | 37.78 | 45.47 | **37.93** | 40.94 | 66.12 | 45.27 |
| NGP | 24.12 | 25.63 | 25.67 | 16.59 | **11.33** | 17.76 | 12.17 | NGP | 44.55 | 73.04 | 60.21 | 51.72 | **43.61** | 59.47 | 36.88 |
| PG | 12.23 | 16.18 | 15.87 | 12.03 | 15.37 | **24.48** | 14.48 | PG | 38.69 | 64.08 | 60.93 | 44.85 | 41.15 | **42.78** | 46.92 |
| SS | 13.55 | 17.65 | 10.55 | 9.34 | 8.49 | 18.75 | **7.15** | SS | 30.22 | 51.23 | 33.48 | 32.41 | 30.57 | 51.83 | **35.48** |

extremely high test RMSEs across all targets (e.g., PG→HL: 64.08, PG→SS: 46.92). Similarly, NGP shows large within-region error (43.61) and high off-diagonal errors, e.g., NGP→HL: 73.04, confirming weak model fit.

Across both crops, the GNN-RNN model shows significantly better cross-region generalization than MMST-ViT. For soybean, average RMSEs for GNN-RNN across test regions are consistently 3–5 points lower than those for MMST-ViT. The corn results reveal even larger disparities, with MMST-ViT frequently exceeding 50 RMSE on cross-region tests (e.g., HL→PG: 89.87, NGP→HL: 73.04), while GNN-RNN maintains more bounded errors.

Several consistent trends emerge. First, HL and NGP are relatively strong source and target regions across both crops, and especially under the GNN-RNN model. Second, PG exhibits poor generalization in both directions, suggesting significant structural dissimilarity. Third, MMST-ViT shows more variance in generalization and higher errors across nearly all region pairs, possibly due to overfitting or insufficient spatial inductive bias. Finally, within-region year-ahead predictions are generally more accurate than cross-region evaluations, though the gap varies by model and crop. Taken together, these results support the use of GNN-RNN for spatio-temporal generalization, particularly with training from HL or NGP, and caution against deploying MMST-ViT in OOD settings without further adaptation or regularization.

## 5.4 Performance in Realistic Deployment Conditions

**Performance Gap Rationale and Calculation.** To assess the practical utility of the models in deployment scenarios, we evaluate how well they generalize across unseen geographies—an essential capability in real-world agricultural settings where labeled data may not be available in all regions. Specifically, we compute the performance gap between two settings: (i) Different region scenario, where the test region is held out during training (e.g., trained in regions A+B for 2017 to 2021, and tested in region C for 2022), and (ii) the same-region year-ahead, where both training and test data belong to the same region but are separated temporally (e.g., trained region A for 2017-2021 and tested in region A for 2022).The performance gap is defined as the relative increase in RMSE (see appendix).

Table 7 summarizes results across three OOD cases as defined in Table 4. GNN-RNN consistently achieves lower absolute RMSE across both corn and soybean predictions, even under OOD settings. This makes it a stronger candidate for deployment where minimizing prediction error is critical. However, the performance gap is more variable for GNN-RNN, particularly in harder OOD

Table 7: OOD vs. same-region RMSE (bu/acre) across crops, models, and scenarios. The scenario cases are detailed in Table 4.

| Crop | Model | Scenario | RMSE(Diff region year-ahead) | RMSE (same-region year-ahead) | Performance Gap (%) |
|---|---|---|---|---|---|
| Soybean | MMST-ViT | Case 1 | 9.04 | 8.93 | 1.23 |
| | | Case 2 | 11.63 | 10.18 | 14.24 |
| | | Case 3 | 12.19 | 11.33 | 7.59 |
| Corn | MMST-ViT | Case 1 | 30.92 | 26.06 | 18.65 |
| | | Case 2 | 34.93 | 33.37 | 4.67 |
| | | Case 3 | 50.26 | 43.61 | 15.25 |
| Soybean | GNN-RNN | Case 1 | 6.92 | 5.46 | 26.75 |
| | | Case 2 | 9.75 | 6.15 | 58.53 |
| | | Case 3 | 11.11 | 7.11 | 56.20 |
| Corn | GNN-RNN | Case 1 | 27.62 | 26.69 | 3.48 |
| | | Case 2 | 27.75 | 22.67 | 22.40 |
| | | Case 3 | 32.07 | 21.65 | 48.13 |

cases—indicating higher sensitivity to distribution shift. MMST-ViT, while slightly less accurate overall, exhibits more stable performance gaps across regions and crops.

## 6 Discussion

**Limitations**   Despite their complementary strengths, both models and the underlying dataset exhibit notable limitations. CropNet repackages publicly available sources (Sentinel-2, HRRR, USDA) without introducing new domain-expert curation. While the dataset does provide NDVI imagery, neither GNN-RNN nor MMST-ViT incorporate this modality in their current implementations. Other widely-used indices such as EVI (Huete et al., 2002) are also absent, and the dataset remains limited to county-level aggregation, precluding field-scale yield estimation.

**Impact of Sentinel-2 bands**   While CropNet is the first terabyte-scale, multi-modal benchmark for county-level yield forecasting, its design choices impose important limitations. Only four of Sentinel-2's twelve spectral bands are provided—omitting red-edge bands crucial for detecting vegetation stress (Krisp and Scheinert, 2021)—and all imagery is Level-1C (un-corrected), leaving atmospheric effects unaddressed (Topping et al., 2019). Furthermore, spatial aggregation to 9 km × 9 km grid cells erases intra-county heterogeneity, precluding field-level analysis.

**Sentinel-2 imagery resolution and uneven grid size**   The number of grids per county is also highly variable, ranging from fewer than 5 to over 130, which introduces unequal spatial sampling across counties and likely contributes to model underperformance by biasing learning toward large, extensively sampled areas. The heterogeneity in grid coverage likely exacerbates cross-region transfer challenges as well, since counties with few grids contribute noisier or less informative features to the learned representations, reducing generalization robustness under distribution shifts.

Higher-resolution approaches like using MODIS (1 km resolution) could mitigate this by enabling more uniform representation of small or fragmented counties. Recent studies have shown that temperature anomalies, more than precipitation variability, are a primary driver of yield variation globally (Iizumi and Sakai, 2020), making the loss of local variability particularly consequential. Additionally, other global datasets such as EarthStat have demonstrated the utility of higher spatial resolution and field-level yield mapping (Monfreda et al., 2008), which CropNet does not currently support.

**Modelling scope**  MMST-ViT does not integrate soil properties or management variables, which are known to substantially influence yield outcomes. In contrast, GNN-RNN has optional support for these factors, though they were not used in the core experiments presented due to data unavailability. The five-year (2017–2021) training window may be insufficient to capture long-term climatic transitions, and the lack of integration with process-based or operational baselines (Lobell et al., 2015) hinders comprehensive assessment. Furthermore, the evaluation omits process-based crop models such as DSSAT (Jones et al., 2001) or APSIM (Robertson et al., 2012), which are frequently used in agricultural forecasting pipelines.

**Generalization Patterns**  Our comprehensive out-of-distribution evaluation reveals that the GNN-RNN model—combining a lightweight GraphSAGE encoder with an LSTM—exhibits the most robust spatial transferability across over 1,200 U.S. counties. Even when deployed in USDA Farm Resource Regions unseen during training, it often maintains reasonable RMSEs and positive correlations, indicating that its learned county embeddings capture both local geographic structure and temporal climate trends. Prior work using deep Gaussian processes for yield prediction also emphasizes the importance of robust uncertainty modeling under spatial shift (You et al., 2017b). By contrast, both GNN-RNN and MMST-ViT suffer severe degradation under spatio-temporal domain shifts (leave-one-region-out evaluations), with frequent instances of negative $R^2$ and inverse correlation. Notably, MMST-ViT's performance implodes on truly unseen geographies—despite strong in-domain accuracy on the four states originally reported (IL, IA, MS, LA)—suggesting it relies heavily on memorized region-specific statistics rather than learning transferrable patterns.

**Poor performance in PG**  The Prairie Gateway (PG) consistently emerges as the hardest out-of-distribution target in our leave-one-cluster-out evaluation, driven by several factors: a) its hot, semi-arid climate with chronically high vapor-pressure deficit induces yield responses not represented in the humid training regions; b) widespread supplemental irrigation, unobserved in HRRR inputs, leads models trained on rain-fed systems to misinterpret precipitation anomalies; c) large internal heterogeneity in crop types, elevation (500–1700 m), and soil properties makes it difficult for a single feature mapping to generalize across PG counties; d) USDA yield series in PG are sparser and noisier, increasing label uncertainty; and e) frequent compound hot–dry extremes and late-season weather hazards violate the conditional independence assumptions embedded in source-region models. The omission of Sentinel-2's red-edge bands, which are particularly sensitive to early vegetation stress and chlorophyll content (Krisp and Scheinert, 2021), may disproportionately impair model performance in semi-arid or drought-prone regions such as Prairie Gateway, where early stress detection is critical for yield outcomes.

**Broader Impact Statement**  The observed regional disparities in generalization performance raise important societal considerations. If predictive models are systematically more reliable in

historically well-resourced, rain-fed regions (e.g., Heartland, Northern Crescent) and less reliable in more vulnerable, semi-arid regions (e.g., Prairie Gateway), there is a risk of exacerbating existing inequalities in agricultural decision support. Regions already facing greater climate risk, variable yields, and water scarcity could be further marginalized by inaccurate forecasts. Therefore, ensuring equitable model performance across diverse agro-climatic zones is not only a technical challenge but also an essential component of promoting fair access to the benefits of AI-driven agricultural innovation.

**Mitigation approaches**    Several avenues exist to mitigate the poor transferability into challenging regions such as PG: a) incorporating additional covariates such as irrigation intensity, satellite-derived evapotranspiration, or groundwater depth to explicitly represent water management; b) using domain-aware normalization schemes that adjust weather feature scaling regionally, reducing feature distribution shift; c) applying domain-adversarial training or feature alignment techniques to harmonize latent representations across climate regimes; d) supplementing training with a small, stratified sample of PG counties to expose the model to the semi-arid, irrigated production regime during learning; and e) combining data-driven predictions with biophysical priors, such as water-limited potential yield envelopes, to maintain physical plausibility under extreme conditions. We note that GNN-RNN model can incorporate soil and management variables.

## 7    Conclusion

This study provides the first systematic evaluation of spatio-temporal out-of-distribution generalization in crop yield prediction, benchmarking state-of-the-art deep learning models under realistic climate-driven distribution shifts. Through comprehensive experiments across 1,200+ U.S. counties using scientifically grounded regional splits, we reveal significant limitations in current approaches that were previously obscured by in-domain evaluation protocols.

Our key findings include: (1) substantial performance degradation under spatial transfer, with models frequently achieving negative $R^2$ values in unseen regions; (2) asymmetric transferability patterns, where certain regions (Prairie Gateway) consistently resist generalization while others (Eastern Uplands) transfer well; (3) architectural differences in robustness, with GNN-RNN demonstrating superior cross-region stability compared to MMST-ViT despite requiring $100\times$ less computational resources; and (4) the critical importance of incorporating spatial relationships, as demonstrated through ablation studies.

These negative results provide valuable insights for the agricultural AI community, highlighting the gap between controlled evaluation settings and real-world deployment challenges. Our evaluation framework, based on USDA Farm Resource Regions, offers a standardized protocol for future robustness assessments in agricultural ML. The consistent failure modes we identify—particularly in semi-arid, irrigated regions—point toward specific research directions for improving model generalization under climate variability.

While deep learning shows promise for improving crop yield forecasting, our findings underscore the necessity of rigorous out-of-distribution evaluation and transparent reporting of generalization limitations. Future work should prioritize domain adaptation techniques and incorporate physical constraints to improve robustness across diverse agro-ecological conditions.

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

## A    Appendix / supplemental material

Table 8: Dataset parameters used in the study

| Dataset | Parameters |
| --- | --- |
| SENTINEL-2 | RGB imagery 9x9 km grids, 40 meters resolution |
| USDA | Production (Corn and Soybean), Yield (bu/acre) |
| HRRR | Averaged Temperature, Maximal Temperature, Minimal Temperature, Precipitation, Relative Humidity, Wind Gust, Wind Speed, Downward Shortwave Radiation Flux, Vapor Pressure Deficit |

The CropNet dataset is publicly available (cited [Lin et al., 2024]) and accessible at `https://huggingface.co/datasets/CropNet/CropNet`. All code used in this study is from open-source repositories, with links provided as footnotes in the main paper: GNN-RNN (`https://github.com/JunwenBai/GNN-RNN`), MMST-ViT (`https://github.com/fudong03/MMST-ViT`).

**Linear baseline using Lasso regression**    We applied the Lasso (Least Absolute Shrinkage and Selection Operator) regression method to predict crop yield using structured inputs derived from weather, soil, and management data. The model was trained on standardized output variables

using only the training data. A grid search was conducted over a predefined range of regularization strengths ($\alpha \in \{0.01, 0.1, 1, 10, 100, 1000, 10000\}$), and each configuration was evaluated across three random seeds to account for training variability. The best-performing configuration was selected based on the average test set performance, evaluated using root mean squared error (RMSE), coefficient of determination ($R^2$), and Pearson correlation coefficient. Finally, predictions were rescaled to the original units.

**GNN-RNN model** The GNN-RNN model is a deep learning framework designed to capture both spatial and temporal dependencies in crop yield prediction. Traditional yield prediction models often assume that counties are independent, ignoring the geographic correlations that influence crop growth. To address this limitation, (Fan et al., 2022) proposed the GNN-RNN architecture. It builds upon the CNN-RNN framework (Khaki et al., 2019) by introducing a graph-based component that refines county-level embeddings using neighboring counties' data before passing them to the LSTM. Experimental results indicate that GNN-RNN outperforms CNN-RNN on multiple yield prediction tasks, highlighting the importance of incorporating geospatial information in deep learning models (Fan et al., 2022). The GNN-RNN model consists of three key components:

- Feature Embedding Extraction: Each county's climatic and soil features are processed separately before being fused into an annual embedding. The model employs a 1D CNN to extract temporal features from weekly weather and land surface data, while a separate CNN processes static soil quality features. These embeddings are concatenated to form a county-year-level representation (Fan et al., 2022).

- Graph Neural Network (GNN) for Spatial Dependencies: The model represents counties as nodes in a graph, where edges connect neighboring counties based on geographic adjacency. A GraphSAGE-based GNN (Hamilton et al., 2017) aggregates information from a county's surrounding areas, allowing local predictions to be influenced by broader regional trends (Fan et al., 2022).The GNN-processed embeddings incorporate spatial dependencies, ensuring that the model captures geographic influences on crop yield.

- Recurrent Neural Network (RNN) for Temporal Dependencies:A Long Short-Term Memory (LSTM) network (Hochreiter and Schmidhuber, 1997) processes multi-year embeddings for each county, modeling long-term climatic effects and historical yield trends (Fan et al., 2022). The final output embedding from the LSTM is used to predict county-level crop yields for the current year.

**MMST-ViT Model** Unlike the GNN-RNN model which relied solely on meteorological and static environmental features MMST-ViT integrates remote sensing data (satellite imagery) and weather-based climate indicators. The use of transformer-based attention mechanisms allows the model to capture both fine-grained spatial dependencies and long-term temporal trends. The MMST-ViT model consists of three key components:

- Remote-Sensing imagery Feature Extraction: A Pyramid Vision Transformer (PVT) (Wang et al., 2021) is used to analyze remote sensing imagery. To mitigate the tendency of transformer-based to overfit, the Pyramid Vision Transformer (PVT) backbone is pretrained using a contrastive learning framework similar to SimCLR (Chen et al., 2020). This

pretraining strategy enhances the model's ability to generalize by learning robust feature representations from unlabeled data.

- Spatial Transformer for Geographic Dependencies:MMST-ViT employs self-attention mechanisms to implicitly learn spatial dependencies. This approach is motivated by (Dosovitskiy et al., 2021), who showed that vision transformers can outperform CNNs by capturing global spatial information without predefined adjacency constraints. The self attention mechanisms allows the model to learn spatial correlations across regions (counties, in this study).

- Temporal Transformer for Climate Trends: To explicitly incorporate climate trends, a Temporal Multi-Head Attention module models long-term climate variability, aligning with recent work emphasizing the growing influence of climate change on agricultural yields (Ortiz-Bobea et al., 2018). Unlike LSTM-based approaches, which have been widely used for time-series forecasting in agriculture (You et al., 2017a), the transformer-based framework allows MMST-ViT to capture long-range dependencies more efficiently (Vaswani et al., 2017).

Table 9: Sensitivity analysis parameters* and model hyperparameters for the GNN-RNN model. Parameters marked with * were varied individually during sensitivity analysis. Bold values indicate the baseline configuration used in the main experiments.

| Hyperparameter | Values Tested or Used |
| --- | --- |
| *Sensitivity Parameters (Ablated)* | |
| Number of GNN Layers* ($n_{\text{layers}}$) | **2**, 3, 4 |
| Aggregation Type* | **pool**, mean, gcn |
| Dropout Rate* | **0**, 0.5, 0.7 |
| *Fixed Hyperparameters* | |
| Temporal Sequence Length ($\Delta t$) | 5 years |
| Learning Rate | 1e–4 |
| Batch Size | 64 |
| LR Scheduling | Cosine ( $T_0 = 100$, $\eta_{\min} = 1e$–6) |
| Epochs | 200 |
| Optimizer | Adam |
| Loss Function | Log-cosh |

## A.1 UMAP weather embeddings

To evaluate the robustness of our models in realistic out-of-distribution (OOD) deployment scenarios, we leverage historical weather data and visualize its structure using UMAP embeddings. Specifically, we extract weekly weather variables from 2017 to 2022 and compute their low-dimensional UMAP representations. Each embedding is then colored according to its USDA Farm Resource Region (FRR) to assess the coherence of the USDA-based clustering.

As shown in Figure 2, the weather embeddings naturally cluster according to USDA Farm Resource Regions, confirming that the USDA FRRs naturally form distinct clusters based on weather patterns.

Table 10: Sensitivity analysis parameters* and model hyperparameters for the GNN-RNN model. Parameters marked with * were varied individually during sensitivity analysis. Bold values indicate the baseline configuration used in the main experiments.

| Hyperparameter | Values Tested or Used |
|---|---|
| *Sensitivity Parameters (Ablated)* | |
| Embedding Dimension* | 128, **512** |
| Dropout Rate* | **0**, 0.3, 0.5 |
| *Shared Training Configuration* | |
| Optimizer | AdamW |
| Weight Decay | 0.05 |
| Learning Rate Schedule | Cosine decay with warmup |
| *Pretraining (Self-Supervised)* | |
| Epochs | 200 |
| Initial Learning Rate | 1e–4 |
| Warmup Epochs | 20 |
| Batch Size | 64 |
| AdamW $(\beta_1, \beta_2)$ | (0.9, 0.95) |
| Augmentation | Random crop, horizontal flip, Gaussian blur, color jitter |
| *Fine-tuning (Supervised)* | |
| Epochs | 100 |
| Initial Learning Rate | 1e–3 |
| Warmup Epochs | 5 |
| AdamW $(\beta_1, \beta_2)$ | (0.9, 0.999) |

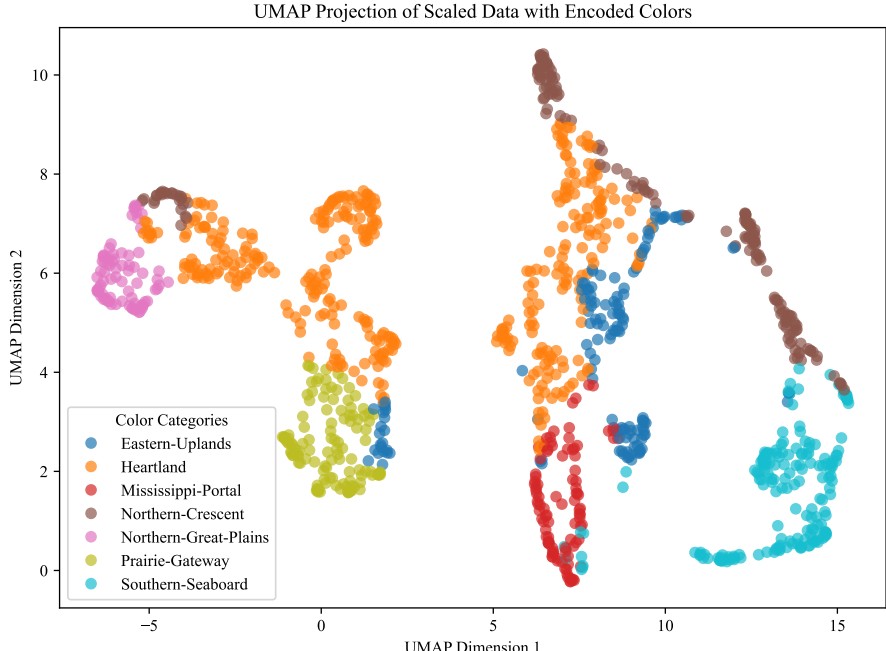

Figure 2: UMAP visualization of weekly weather embeddings (2017-2022), colored by USDA Farm Resource Regions. The clustering structure confirms that USDA FRRs provide a meaningful partitioning of agricultural regions based on weather characteristics.

This provides further evidence that USDA FRRs are a meaningful and scientifically grounded way to partition the data for OOD evaluation. To systematically evaluate model generalization under varying levels of distribution shift, we define three out-of-distribution (OOD) scenarios—Case 1, Case 2, and Case 3—based on a qualitative synthesis of multiple indicators. Specifically, we consider: (i) the distribution of clusters in UMAP space, (ii) their geographical proximity, and (iii) performance patterns observed in leave-cluster-out (LCO) cross-validation. While the cases are not explicitly labeled by difficulty, we hypothesize that Case 1 likely corresponds to an easier distribution shift, involving test regions that are closely aligned with the training distribution in both UMAP space and geography. Case 2 appears to reflect a moderate shift, with partial separation in UMAP space and moderate geographic distance. Case 3 seems to represent the most challenging scenario, involving substantial separation in UMAP space, geographic remoteness, and typically larger performance drops in LCO validation.

Table 11: Performance metrics for Leave-One-Region-Out year-ahead evaluation for soybean yield prediction using GNN-RNN model. [l,d,type] corresponds to number of layers, dropout and aggregation type of the graph convolutional network.

| Method | Excl Reg | RMSE Incl | R² Incl | Corr Incl | RMSE Excl | R² Excl | Corr Excl |
|---|---|---|---|---|---|---|---|
| Lasso baseline | PG | 10.90 ± 0.43 | 0.02 ± 0.03 | 0.69 ± 0.05 | 17.03 ± 1.27 | -0.11 ± 0.04 | 0.19 ± 0.06 |
| | EU | 11.44 ± 0.62 | 0.14 ± 0.04 | 0.64 ± 0.06 | 10.90 ± 0.87 | 0.14 ± 0.04 | 0.67 ± 0.05 |
| | HL | 10.58 ± 0.82 | 0.12 ± 0.03 | 0.52 ± 0.07 | 15.87 ± 1.28 | -2.47 ± 0.15 | 0.31 ± 0.03 |
| | MSP | 10.41 ± 0.71 | 0.32 ± 0.05 | 0.63 ± 0.06 | 12.08 ± 1.07 | -1.35 ± 0.12 | -0.15 ± 0.02 |
| | NC | 12.39 ± 0.91 | 0.07 ± 0.03 | 0.60 ± 0.05 | 8.17 ± 0.45 | 0.10 ± 0.03 | 0.44 ± 0.03 |
| | NGP | 11.08 ± 0.77 | 0.18 ± 0.04 | 0.60 ± 0.06 | 11.32 ± 1.14 | -1.42 ± 0.13 | 0.04 ± 0.00 |
| | SS | 10.86 ± 0.68 | 0.26 ± 0.05 | 0.61 ± 0.06 | 10.62 ± 0.95 | -1.06 ± 0.10 | -0.04 ± 0.00 |
| l=2,d=0 ,pool | EU | 8.60 ± 0.42 | 0.52 ± 0.05 | 0.73 ± 0.05 | 6.96 ± 0.81 | 0.65 ± 0.05 | 0.84 ± 0.05 |
| | HL | 9.60 ± 0.65 | 0.28 ± 0.04 | 0.56 ± 0.06 | 8.31 ± 0.76 | 0.05 ± 0.03 | 0.55 ± 0.06 |
| | MSP | 8.33 ± 0.51 | 0.57 ± 0.05 | 0.76 ± 0.05 | 10.22 ± 0.83 | -0.68 ± 0.08 | -0.10 ± 0.00 |
| | NC | 9.07 ± 0.58 | 0.50 ± 0.05 | 0.73 ± 0.05 | 10.28 ± 0.91 | -0.42 ± 0.06 | -0.18 ± 0.00 |
| | NGP | 8.77 ± 0.63 | 0.48 ± 0.05 | 0.72 ± 0.06 | 9.53 ± 0.84 | -0.71 ± 0.09 | -0.26 ± 0.01 |
| | PG | 7.06 ± 0.45 | 0.59 ± 0.06 | 0.78 ± 0.05 | 23.26 ± 2.18 | -0.56 ± 0.05 | 0.32 ± 0.01 |
| | SS | 8.88 ± 0.57 | 0.51 ± 0.05 | 0.73 ± 0.05 | 10.83 ± 0.99 | -0.64 ± 0.06 | -0.39 ± 0.01 |
| l=2,d=0 ,mean | EU | 8.74 ± 0.53 | 0.50 ± 0.05 | 0.72 ± 0.06 | 5.97 ± 0.62 | 0.74 ± 0.05 | 0.86 ± 0.05 |
| | HL | 9.08 ± 0.48 | 0.36 ± 0.04 | 0.60 ± 0.06 | 7.94 ± 0.81 | 0.13 ± 0.04 | 0.58 ± 0.06 |
| | MSP | 7.90 ± 0.47 | 0.61 ± 0.06 | 0.79 ± 0.05 | 9.37 ± 0.75 | -0.41 ± 0.07 | -0.07 ± 0.00 |
| | NC | 9.16 ± 0.70 | 0.49 ± 0.05 | 0.71 ± 0.06 | 9.43 ± 0.50 | -0.20 ± 0.05 | 0.06 ± 0.00 |
| | NGP | 8.26 ± 0.73 | 0.54 ± 0.05 | 0.74 ± 0.06 | 7.51 ± 0.62 | -0.06 ± 0.04 | 0.29 ± 0.01 |
| | PG | 7.04 ± 0.46 | 0.59 ± 0.06 | 0.77 ± 0.05 | 23.57 ± 2.25 | -1.12 ± 0.07 | 0.27 ± 0.02 |
| | SS | 8.98 ± 0.67 | 0.50 ± 0.05 | 0.72 ± 0.06 | 11.78 ± 1.14 | -1.53 ± 0.07 | -0.39 ± 0.01 |
| l=2,d=0 ,gcn | EU | 9.09 ± 0.63 | 0.46 ± 0.05 | 0.69 ± 0.06 | 6.75 ± 0.54 | 0.67 ± 0.05 | 0.83 ± 0.06 |
| | HL | 8.90 ± 0.71 | 0.38 ± 0.04 | 0.63 ± 0.06 | 7.74 ± 0.58 | 0.17 ± 0.04 | 0.64 ± 0.05 |
| | MSP | 8.43 ± 0.55 | 0.56 ± 0.05 | 0.75 ± 0.06 | 9.39 ± 0.93 | -0.42 ± 0.07 | -0.00 ± 0.00 |
| | NC | 8.93 ± 0.49 | 0.52 ± 0.05 | 0.72 ± 0.06 | 10.05 ± 1.10 | -0.36 ± 0.03 | -0.13 ± 0.00 |
| | NGP | 9.00 ± 0.52 | 0.46 ± 0.05 | 0.71 ± 0.06 | 8.98 ± 0.83 | -0.52 ± 0.04 | -0.03 ± 0.00 |
| | PG | 7.25 ± 0.65 | 0.57 ± 0.06 | 0.77 ± 0.05 | 25.33 ± 1.86 | -1.45 ± 0.07 | 0.07 ± 0.00 |
| | SS | 8.08 ± 0.66 | 0.59 ± 0.06 | 0.78 ± 0.05 | 11.15 ± 1.22 | -1.27 ± 0.06 | -0.39 ± 0.01 |
| l=3,d=0 ,pool | EU | 9.29 ± 0.57 | 0.44 ± 0.05 | 0.71 ± 0.06 | 7.72 ± 0.75 | 0.57 ± 0.05 | 0.83 ± 0.06 |
| | HL | 9.88 ± 0.83 | 0.24 ± 0.04 | 0.54 ± 0.06 | 8.58 ± 0.84 | -0.02 ± 0.03 | 0.56 ± 0.03 |
| | MSP | 8.26 ± 0.74 | 0.57 ± 0.05 | 0.76 ± 0.06 | 10.63 ± 1.08 | -0.82 ± 0.10 | -0.02 ± 0.00 |
| | NC | 8.63 ± 0.60 | 0.55 ± 0.05 | 0.74 ± 0.06 | 9.55 ± 0.96 | -0.23 ± 0.06 | 0.09 ± 0.00 |
| | NGP | 8.07 ± 0.51 | 0.56 ± 0.05 | 0.76 ± 0.06 | 6.38 ± 0.58 | 0.23 ± 0.05 | 0.56 ± 0.03 |
| | PG | 7.57 ± 0.49 | 0.53 ± 0.05 | 0.74 ± 0.06 | 24.20 ± 2.31 | -1.23 ± 0.07 | 0.25 ± 0.02 |
| | SS | 8.65 ± 0.56 | 0.53 ± 0.05 | 0.73 ± 0.06 | 11.36 ± 1.15 | -1.35 ± 0.09 | -0.43 ± 0.03 |
| l=4,d=0 ,pool | EU | 8.91 ± 0.67 | 0.48 ± 0.05 | 0.71 ± 0.06 | 6.36 ± 0.49 | 0.71 ± 0.05 | 0.85 ± 0.05 |
| | HL | 9.53 ± 0.65 | 0.29 ± 0.04 | 0.58 ± 0.06 | 8.07 ± 0.90 | 0.10 ± 0.04 | 0.59 ± 0.06 |
| | MSP | 8.15 ± 0.49 | 0.58 ± 0.06 | 0.78 ± 0.05 | 10.45 ± 1.10 | -0.76 ± 0.08 | -0.10 ± 0.00 |
| | NC | 9.12 ± 0.66 | 0.49 ± 0.05 | 0.74 ± 0.06 | 9.77 ± 0.85 | -0.29 ± 0.07 | -0.03 ± 0.00 |
| | NGP | 8.20 ± 0.59 | 0.55 ± 0.05 | 0.77 ± 0.06 | 6.29 ± 0.63 | 0.25 ± 0.05 | 0.57 ± 0.06 |
| | PG | 7.49 ± 0.77 | 0.54 ± 0.05 | 0.75 ± 0.06 | 24.99 ± 2.13 | -1.38 ± 0.09 | 0.31 ± 0.02 |

*Continued from previous page*

|  |  | RMSE Incl | R² Incl | Corr Incl | RMSE Excl | R² Excl | Corr Excl |
|---|---|---|---|---|---|---|---|
|  | SS | 8.95 ± 0.64 | 0.50 ± 0.05 | 0.71 ± 0.06 | 12.07 ± 1.22 | -1.66 ± 0.07 | -0.41 ± 0.02 |
| l=2,d=0.5,pool | EU | 8.35 ± 0.63 | 0.55 ± 0.05 | 0.74 ± 0.06 | 6.33 ± 0.53 | 0.71 ± 0.05 | 0.85 ± 0.06 |
|  | HL | 9.83 ± 0.71 | 0.24 ± 0.04 | 0.54 ± 0.06 | 8.47 ± 0.90 | 0.01 ± 0.03 | 0.54 ± 0.06 |
|  | MSP | 9.24 ± 0.51 | 0.47 ± 0.05 | 0.69 ± 0.06 | 9.38 ± 0.91 | -0.42 ± 0.08 | 0.10 ± 0.00 |
|  | NC | 8.95 ± 0.62 | 0.51 ± 0.05 | 0.72 ± 0.06 | 10.26 ± 1.01 | -0.42 ± 0.07 | -0.06 ± 0.00 |
|  | NGP | 8.58 ± 0.56 | 0.51 ± 0.05 | 0.72 ± 0.06 | 8.70 ± 0.81 | -0.43 ± 0.08 | 0.26 ± 0.00 |
|  | PG | 7.65 ± 0.64 | 0.52 ± 0.05 | 0.73 ± 0.06 | 23.08 ± 2.25 | -1.03 ± 0.08 | 0.25 ± 0.01 |
|  | SS | 8.09 ± 0.71 | 0.59 ± 0.06 | 0.77 ± 0.05 | 11.39 ± 1.26 | -1.37 ± 0.07 | -0.35 ± 0.01 |
| l=2,d=0.7,pool | EU | 8.81 ± 0.73 | 0.49 ± 0.05 | 0.70 ± 0.06 | 6.49 ± 0.57 | 0.69 ± 0.05 | 0.84 ± 0.06 |
|  | HL | 9.61 ± 0.55 | 0.28 ± 0.04 | 0.55 ± 0.06 | 8.08 ± 0.68 | 0.10 ± 0.04 | 0.53 ± 0.06 |
|  | MSP | 8.65 ± 0.66 | 0.53 ± 0.05 | 0.74 ± 0.06 | 9.61 ± 1.06 | -0.49 ± 0.06 | 0.01 ± 0.00 |
|  | NC | 9.48 ± 0.55 | 0.46 ± 0.05 | 0.75 ± 0.06 | 10.97 ± 1.28 | -0.62 ± 0.07 | -0.03 ± 0.00 |
|  | NGP | 9.67 ± 0.75 | 0.37 ± 0.04 | 0.66 ± 0.06 | 10.88 ± 0.96 | -1.23 ± 0.05 | -0.15 ± 0.00 |
|  | PG | 7.27 ± 0.60 | 0.56 ± 0.06 | 0.76 ± 0.05 | 24.01 ± 2.33 | -1.20 ± 0.06 | 0.18 ± 0.00 |
|  | SS | 7.94 ± 0.53 | 0.61 ± 0.06 | 0.79 ± 0.05 | 10.54 ± 1.20 | -1.03 ± 0.07 | -0.29 ± 0.01 |
| Ablation CNN-RNN | EU | 9.93 ± 0.71 | 0.36 ± 0.04 | 0.63 ± 0.06 | 7.59 ± 0.65 | 0.58 ± 0.05 | 0.84 ± 0.06 |
|  | HL | 9.06 ± 0.67 | 0.36 ± 0.04 | 0.66 ± 0.06 | 8.81 ± 0.74 | -0.07 ± 0.02 | 0.71 ± 0.04 |
|  | MSP | 9.33 ± 0.64 | 0.46 ± 0.05 | 0.71 ± 0.06 | 9.59 ± 0.89 | -0.48 ± 0.03 | 0.14 ± 0.01 |
|  | NC | 10.23 ± 0.75 | 0.37 ± 0.04 | 0.66 ± 0.06 | 9.07 ± 0.91 | -0.11 ± 0.01 | 0.14 ± 0.00 |
|  | NGP | 9.00 ± 0.68 | 0.46 ± 0.05 | 0.70 ± 0.06 | 8.11 ± 0.79 | -0.24 ± 0.01 | 0.47 ± 0.02 |
|  | PG | 10.23 ± 0.77 | 0.14 ± 0.04 | 0.57 ± 0.07 | 19.41 ± 2.17 | -0.44 ± 0.01 | 0.19 ± 0.00 |
|  | SS | 7.98 ± 0.59 | 0.60 ± 0.06 | 0.79 ± 0.05 | 10.20 ± 1.15 | -0.90 ± 0.02 | -0.18 ± 0.00 |

Table 12: Performance metrics for Leave-One-Region-Out year-ahead evaluation for soybean yield prediction using MMST-ViT model. [emb,drop] corresponds to the embedding dimension and dropout of the model.

| Method | Excl Reg | RMSE Incl | R² Incl | Corr Incl | RMSE Excl | R² Excl | Corr Excl |
|---|---|---|---|---|---|---|---|
| Lasso baseline | PG | 10.90 ± 0.43 | 0.02 ± 0.03 | 0.69 ± 0.05 | 17.03 ± 1.27 | -0.11 ± 0.04 | 0.19 ± 0.06 |
|  | EU | 11.44 ± 0.62 | 0.14 ± 0.04 | 0.64 ± 0.06 | 10.90 ± 0.87 | 0.14 ± 0.02 | 0.67 ± 0.05 |
|  | HL | 10.58 ± 0.82 | 0.12 ± 0.03 | 0.52 ± 0.06 | 15.87 ± 1.28 | -2.47 ± 0.15 | 0.31 ± 0.01 |
|  | MSP | 10.41 ± 0.71 | 0.32 ± 0.05 | 0.63 ± 0.06 | 12.08 ± 1.07 | -1.35 ± 0.12 | -0.15 ± 0.00 |
|  | NC | 12.39 ± 0.91 | 0.07 ± 0.03 | 0.60 ± 0.05 | 8.17 ± 0.45 | 0.10 ± 0.00 | 0.44 ± 0.01 |
|  | NGP | 11.08 ± 0.77 | 0.18 ± 0.04 | 0.60 ± 0.06 | 11.32 ± 1.14 | -1.42 ± 0.13 | 0.04 ± 0.00 |
|  | SS | 10.86 ± 0.68 | 0.26 ± 0.05 | 0.61 ± 0.06 | 10.62 ± 0.95 | -1.06 ± 0.08 | -0.04 ± 0.00 |
| e=512, drop=0 | EU | 13.06 ± 0.47 | 0.48 ± 0.05 | 0.69 ± 0.06 | 9.16 ± 0.42 | 0.62 ± 0.06 | 0.79 ± 0.05 |
|  | HL | 13.46 ± 0.61 | 0.30 ± 0.04 | 0.55 ± 0.06 | 11.67 ± 0.53 | 0.10 ± 0.00 | 0.32 ± 0.02 |
|  | MSP | 12.76 ± 0.49 | 0.52 ± 0.06 | 0.72 ± 0.05 | 12.71 ± 0.58 | -0.41 ± 0.03 | -0.09 ± 0.00 |
|  | NC | 13.32 ± 0.57 | 0.46 ± 0.05 | 0.68 ± 0.06 | 9.46 ± 0.39 | 0.65 ± 0.04 | 0.81 ± 0.05 |
|  | NGP | 12.83 ± 0.48 | 0.41 ± 0.05 | 0.64 ± 0.06 | 11.64 ± 0.51 | -0.31 ± 0.04 | -0.14 ± 0.00 |
|  | PG | 11.58 ± 0.43 | 0.57 ± 0.06 | 0.76 ± 0.05 | 20.96 ± 0.91 | -0.99 ± 0.06 | 0.28 ± 0.00 |
|  | SS | 12.49 ± 0.55 | 0.50 ± 0.06 | 0.71 ± 0.06 | 15.32 ± 0.67 | -1.24 ± 0.04 | -0.37 ± 0.00 |

*Continued on next page*

*Continued from previous page*

| | | | | | | | |
|---|---|---|---|---|---|---|---|
| e=512, drop=0.3 | EU | 14.99 ± 0.68 | 0.42 ± 0.05 | 0.65 ± 0.06 | 11.58 ± 0.52 | 0.59 ± 0.06 | 0.77 ± 0.06 |
| | HL | 17.18 ± 0.77 | 0.23 ± 0.04 | 0.48 ± 0.06 | 10.19 ± 0.47 | 0.09 ± 0.04 | 0.30 ± 0.01 |
| | MSP | 13.49 ± 0.62 | 0.49 ± 0.06 | 0.70 ± 0.06 | 24.07 ± 1.05 | -1.35 ± 0.15 | -0.21 ± 0.01 |
| | NC | 15.04 ± 0.66 | 0.40 ± 0.05 | 0.63 ± 0.06 | 13.00 ± 0.59 | -0.29 ± 0.06 | -0.12 ± 0.00 |
| | NGP | 14.72 ± 0.63 | 0.38 ± 0.05 | 0.62 ± 0.06 | 14.66 ± 0.64 | -0.41 ± 0.07 | -0.16 ± 0.00 |
| | PG | 13.48 ± 0.60 | 0.50 ± 0.06 | 0.71 ± 0.06 | 23.44 ± 1.01 | -1.12 ± 0.13 | 0.27 ± 0.00 |
| | SS | 14.75 ± 0.65 | 0.46 ± 0.06 | 0.68 ± 0.06 | 14.29 ± 0.63 | -0.99 ± 0.12 | -0.32 ± 0.01 |
| e=128, drop=0 | EU | 14.01 ± 0.63 | 0.45 ± 0.06 | 0.67 ± 0.06 | 12.59 ± 0.57 | 0.60 ± 0.07 | 0.78 ± 0.06 |
| | HL | 13.95 ± 0.60 | 0.31 ± 0.04 | 0.56 ± 0.06 | 13.82 ± 0.62 | -0.12 ± 0.05 | -0.10 ± 0.00 |
| | MSP | 13.55 ± 0.58 | 0.50 ± 0.06 | 0.71 ± 0.06 | 17.04 ± 0.76 | -0.79 ± 0.11 | -0.15 ± 0.00 |
| | NC | 14.08 ± 0.61 | 0.43 ± 0.05 | 0.66 ± 0.06 | 12.89 ± 0.58 | 0.65 ± 0.07 | 0.81 ± 0.06 |
| | NGP | 14.17 ± 0.59 | 0.40 ± 0.05 | 0.63 ± 0.06 | 9.20 ± 0.42 | 0.41 ± 0.05 | 0.64 ± 0.05 |
| | PG | 13.63 ± 0.62 | 0.52 ± 0.06 | 0.72 ± 0.06 | 16.16 ± 0.72 | -0.85 ± 0.12 | 0.20 ± 0.00 |
| | SS | 13.85 ± 0.60 | 0.48 ± 0.06 | 0.69 ± 0.06 | 14.25 ± 0.64 | -0.99 ± 0.13 | -0.32 ± 0.00 |
| e=128, drop=0.3 | EU | 13.41 ± 0.59 | 0.47 ± 0.06 | 0.68 ± 0.06 | 12.33 ± 0.56 | 0.61 ± 0.07 | 0.78 ± 0.06 |
| | HL | 13.34 ± 0.58 | 0.32 ± 0.04 | 0.57 ± 0.06 | 13.27 ± 0.60 | -0.15 ± 0.05 | -0.11 ± 0.00 |
| | MSP | 13.32 ± 0.57 | 0.51 ± 0.06 | 0.72 ± 0.06 | 13.74 ± 0.41 | -0.41 ± 0.07 | -0.09 ± 0.00 |
| | NC | 13.53 ± 0.60 | 0.44 ± 0.05 | 0.66 ± 0.06 | 12.17 ± 0.55 | 0.69 ± 0.07 | 0.83 ± 0.06 |
| | NGP | 13.48 ± 0.59 | 0.41 ± 0.05 | 0.64 ± 0.06 | 10.63 ± 0.48 | 0.30 ± 0.05 | 0.55 ± 0.04 |
| | PG | 12.63 ± 0.56 | 0.54 ± 0.06 | 0.74 ± 0.06 | 18.61 ± 0.83 | -1.01 ± 0.13 | 0.23 ± 0.00 |
| | SS | 13.47 ± 0.61 | 0.49 ± 0.06 | 0.70 ± 0.06 | 11.41 ± 0.52 | 0.65 ± 0.07 | 0.81 ± 0.05 |
| e=128, drop=0.5 | EU | 16.03 ± 0.72 | 0.40 ± 0.05 | 0.63 ± 0.06 | 14.75 ± 0.66 | 0.59 ± 0.06 | 0.77 ± 0.06 |
| | HL | 18.58 ± 0.83 | 0.20 ± 0.04 | 0.45 ± 0.06 | 11.03 ± 0.50 | 0.09 ± 0.04 | 0.30 ± 0.01 |
| | MSP | 16.16 ± 0.73 | 0.47 ± 0.06 | 0.68 ± 0.06 | 13.01 ± 0.59 | -0.41 ± 0.07 | -0.09 ± 0.00 |
| | NC | 15.67 ± 0.71 | 0.41 ± 0.05 | 0.64 ± 0.06 | 17.10 ± 0.77 | -0.79 ± 0.12 | -0.15 ± 0.00 |
| | NGP | 16.00 ± 0.72 | 0.38 ± 0.05 | 0.62 ± 0.06 | 14.61 ± 0.66 | -0.41 ± 0.07 | -0.16 ± 0.00 |
| | PG | 14.29 ± 0.65 | 0.51 ± 0.06 | 0.72 ± 0.06 | 26.92 ± 1.21 | -1.35 ± 0.15 | 0.21 ± 0.00 |
| | SS | 15.36 ± 0.69 | 0.43 ± 0.06 | 0.66 ± 0.06 | 21.10 ± 0.95 | -1.12 ± 0.13 | -0.32 ± 0.01 |
| Ablation HRRR-only | EU | 15.30 ± 0.67 | 0.32 ± 0.04 | 0.57 ± 0.06 | 11.50 ± 0.52 | 0.54 ± 0.07 | 0.74 ± 0.06 |
| | HL | 14.92 ± 0.63 | 0.28 ± 0.04 | 0.53 ± 0.06 | 12.75 ± 0.58 | -0.12 ± 0.05 | -0.08 ± 0.00 |
| | MSP | 15.67 ± 0.71 | 0.41 ± 0.05 | 0.64 ± 0.06 | 14.03 ± 0.63 | -0.38 ± 0.07 | -0.09 ± 0.00 |
| | NC | 15.11 ± 0.68 | 0.36 ± 0.05 | 0.60 ± 0.06 | 11.63 ± 0.53 | -0.62 ± 0.07 | -0.54 ± 0.02 |
| | NGP | 15.18 ± 0.69 | 0.34 ± 0.05 | 0.58 ± 0.06 | 8.82 ± 0.41 | -0.25 ± 0.05 | -0.23 ± 0.00 |
| | PG | 14.00 ± 0.61 | 0.44 ± 0.05 | 0.66 ± 0.06 | 25.71 ± 1.17 | -0.88 ± 0.12 | -0.19 ± 0.00 |
| | SS | 15.21 ± 0.69 | 0.39 ± 0.05 | 0.62 ± 0.06 | 13.60 ± 0.62 | -0.58 ± 0.06 | -0.22 ± 0.00 |

## OOD Scenario Definition and Performance Gap Metric

We define three out-of-distribution (OOD) scenarios using USDA Farm Resource Regions to evaluate spatiotemporal generalization. These scenarios are heuristically designed based on geographic separation and UMAP-based latent space dissimilarity. Each case involves training on crop yield data from 2017–2021 within selected regions and evaluating on an unseen region in 2022. To quantify the degradation in performance under distribution shift, we define a normalized performance gap metric based on root mean squared error (RMSE):

- **RMSE$_{\text{OOD}}$**: Model trained on 2017–2021 data from a given *Train Region*, evaluated on 2022 data from a disjoint *Test Region*.

- **RMSE$_{\text{IID}}$**: Model trained on 2017–2021 data and evaluated on 2022 data from the *same region* used during training.

The relative performance degradation is computed as:

$$\text{Performance Gap } (\%) = \frac{\text{RMSE}_{\text{OOD}} - \text{RMSE}_{\text{IID}}}{\text{RMSE}_{\text{IID}}} \times 100$$

This normalized performance gap provides a fair comparison of model robustness across regions with inherently different data complexity or signal quality. It adjusts for baseline model performance in each region, removing confounding effects due to region-specific difficulty (e.g., climatic variability, sparse observations, or noisy yield labels). By anchoring performance to an in-distribution baseline, this metric isolates the effect of distribution shift itself, making it particularly well-suited for evaluating OOD generalization in real-world, spatially heterogeneous dataset application such as crop yield prediction.

