# OpenReview forum: "A Systematic Evaluation of Out-of-Distribution Generalization in Crop Yield Prediction"
_TMLR — Accepted by TMLR_

### Review · Reviewer_DhdB · 2025-10-29

**Summary Of Contributions:**

This work evaluates two deep learning architectures for crop yield prediction in out-of-distribution settings. They find limited robustness of the two methods, especially if models are trained only on few data from few geographic areas and then applied to other geographic areas.

**Audience:**

Yes

**Audience Explanation:**

This paper evaluates two methods which have been previously published at ML venues (AAAI & ICCV). It "systematically studies the robustness or generalizability of the published methods" and is thus within scope of TMLR.

**Broader Impact Concerns:**

Broader impact statement is OK.

**Claims And Evidence:**

Yes

**Claims Explanation:**

The work runs extensive experiments to assess the performance of the two methods and ensure their robustness to distribution shifts.

One minor problem was that the authors were unable to reproduce the results reported in the MMST-ViT paper. I would suggest the authors to reach out to MMST-ViT authors and ask for additional information to reproduce the original results. However, given the extensive description i trust the authors of this work to have put significant work into trying to reproduce MMST-ViT results, albeit so far unsuccessfully.

**Requested Changes:**

1. Remove "climate-aware" from the title. This work has little to nothing to do with climate change, so using this term is misleading.
2. Change the classification of page 2: it makes little sense to classify methods as to either use remote sensing or meteorology, as those two are rather complementary types of features and should thus be used simultaneously.
3. I don't get Table 1. Perhaps it should be streamlined to match the storyline and particularly make clear why all the experiments are necessary to have a robust OOD evaluation -- and not just state what the experiments do.
4. Table 5: add the LOO results to this table as an additional row: how good are the models, if trained on all other regions except for the test region?
5. Table 5 & 6: you need to give the colorscale
6. Phrase some sentences more carefully throughout the work, e.g. "While deep learning shows promise for climate-resilient agriculture" in the conclusion, seems a bit strong wording: crop yield prediction is not really the same as "climate-resilient agriculture".

---

### Review · Reviewer_fVyG · 2025-11-02

**Summary Of Contributions:**

The paper is an evaluation of how the two models GNN-RNN and MMST-VIT perform under temporal and spatio-temporal OOD in the CropNet dataset containing data from 1200 US counties split into 7 clusters.

### Strength:
1. The paper’s motivation to examine how crop yield prediction models are affected by OOD shifts across time and space is relevant.
2. Designing three different OOD difficulty scenarios based on UMAP in Table 4 is a good starting point.

### Weakness:
1. The paper is structured as a sequence of reproduction studies, ablation studies, and discussions on the limitations of existing models and the dataset itself, but lacks a coherent narrative flow. Some experiments do not appear directly relevant to the stated motivation of investigating temporal and spatio-temporal ood generalization.
2. This lack of flow stems partly from missing or unclear references between sections. For example, Tables 5 and 6 appear in the main content but are never explicitly referenced. Section 5.2 claims to analyze temporal OOD performance in terms of R², yet none of the tables in the main content seem aligned with this discussion.
3. The purpose of conducting a cross-region transferability study for each region is unclear, given that the paper’s stated goal is to study OOD behavior. Why not instead use the region groupings introduced in Table 4?
4. The discussion section should focus more specifically on identifying which aspects of the model or dataset cause the observed lack of generalization under OOD settings.
5. Using only RMSE and reporting its degradation without any confidence analysis weakens the qualitative findings.

**Audience:**

Yes

**Audience Explanation:**

refer strength

**Claims And Evidence:**

No

**Claims Explanation:**

refer weakness

**Requested Changes:**

refer weakness

---

### Review · Reviewer_g7ua · 2025-12-01

**Summary Of Contributions:**

This paper presents a systematic evaluation of out-of-distribution generalization in crop yield prediction models under climate variability. The authors benchmark two deep learning architectures (GNN-RNN and MMST-ViT) using corn and soybean data from over 1,200 U.S. counties spanning 2017-2022 (the CropNet Dataset), employing USDA Farm Resource Regions as a basis for defining spatial clusters and OOD scenarios.
The main contributions of the paper are: (1) demonstrating significant performance degradation under spatial distribution shift; (2) identifying asymmetric transferability patterns; (3) showing GNN-RNN achieves superior cross-region stability compared to MMST-ViT despite requiring substantially less computational resources; and (4) establishing a standardized evaluation framework for agricultural ML.

I find that the research question considers an important setting for model robustness: realistic spatio-temporal distribution shifts. The use of USDA Regions provides a grounded regional partitioning, and the transparent reporting of negative results is valuable.

Nonetheless the work presents some key weaknesses. The presentation lacks clarity in dataset composition, model adaptation procedures, and experimental setup. Missing explicit citations and insufficient methodological detail make the paper difficult to follow.

**Audience:**

Yes

**Audience Explanation:**

The research addresses a specific case of OOD generalisation in spatiotemporal settings on real data.
The negative results (models achieving negative R2 under certain transfers) could provide valuable insights about failure modes that are often underreported (but they need to be explored more in detail).
However, the current presentation significantly limits accessibility. The paper requires substantial revision to clarify the experimental setup, dataset details, and model adaptation procedures before acceptance.

**Broader Impact Concerns:**

The paper includes an appropriate Broader Impact Statement discussing how regional disparities in model performance could exacerbate existing inequalities in agricultural decision support.
No additional ethical concerns beyond those identified by the authors.

**Claims And Evidence:**

No

**Claims Explanation:**

While the paper presents extensive experimental results, several clarity issues prevent full assessment of the claims:
The dataset description in Section 3 is insufficient. It remains unclear whether Sentinel-2 imagery constitutes a separate dataset requiring integration with CropNet, or whether it is already included. The relationship between weather data, USDA yield data, and Sentinel-2 imagery needs explicit clarification.
Section 5.1 states "we first reproduced the GNN-RNN model using the same architecture and hyperparameters as reported in the original paper" without explicitly citing which paper at that specific point. While Fan et al. 2022 appears later, the immediate context lacks this citation.
The experimental setup lacks critical details about model-to-dataset adaptation. How is the GNN-RNN graph structure defined? How are Sentinel-2 images processed and integrated with weather variables in MMST-ViT? The paper jumps to results without adequately describing these procedures.

**Requested Changes:**

Critical Changes (required for acceptance):

1. Clarify dataset composition and model integration: There is a value in explaining the dataset and which attributes it contains. Describe the data pipeline: preprocessing steps, spatial and temporal alignment, and final input format for each model.
For GNN-RNN, describe how the graph structure is constructed—what defines an edge, typical neighbor count, and handling of isolated counties. For MMST-ViT, describe how the Sentinel-2 patches are matched to counties (given variable grid counts per county), how the four spectral bands are processed, and how imagery features are fused with HRRR weather variables.
2. Add explicit citations throughout Section 5.1: When referencing "the original paper" for GNN-RNN, immediately cite Fan et al. 2022. Similarly, cite Lin et al. 2023 when discussing MMST-ViT reproduction.
3. Expand Section 4: For a practical paper the authors should provide a clear description of: (a) exact train/validation/test splits, (b) hyperparameter selection procedure, (c) evaluation metrics computation, (d) number of random seeds, and (e) computational requirements. Consider adding a figure illustrating the experimental pipeline. Best of all would be to share a code to reproduce the result of the paper.
4. Add a baseline experiment training on all regions: Include an experiment where models train on all seven USDA regions (2017-2021) and test on year 2022 across all regions. This establishes a baseline when no spatial shift is present and quantifies the cost of spatial generalization.

While reading the work I also think that this work would improve with some baselines and additional explorations

5. Provide feature importance analysis: Identify which input features (weather variables, soil properties, spectral bands) are most predictive through ablation studies, attention weight visualization, or permutation importance. This would explain why certain regions transfer poorly and provide actionable insights.
6. Quantify degree of distribution shift: Beyond categorical USDA regions, provide quantitative measures of how "out-of-distribution" each test region is using statistical distance metrics (MMD, Wasserstein distance) on weather features or UMAP embedding distances. Correlate these with performance degradation to understand whether performance scales predictably with shift magnitude.
8. What are the main reasons why Prairie Gateway specifically fails? Can the authors comment a bit on this?

---

### Comment · Reviewer_Pt4L · 2025-11-02

Summarization:
The paper presents the first large-scale study assessing how deep learning models generalize across spatial and temporal domains in crop yield forecasting under climate variability. Using data from over 1,200 U.S. counties and two representative models (GNN-RNN and MMST-ViT), the authors systematically evaluate performance under year-ahead and region-held-out conditions. Results show that both models experience severe degradation under out-of-distribution (OOD) shifts, with GNN-RNN demonstrating better spatial generalization and efficiency than MMST-ViT. The study reveals asymmetric transferability patterns across regions, highlights the Prairie Gateway as particularly challenging, and offers insights for developing more robust, climate-resilient agricultural AI systems.


Strengthens:
1. The paper provides the first systematic benchmark for assessing out-of-distribution (OOD) generalization in crop yield prediction, covering both spatial and temporal shifts across diverse U.S. regions.

2. By focusing on climate-aware prediction and real-world agricultural datasets, the study directly addresses a critical challenge in sustainable food production and climate adaptation.

3. The analysis reveals asymmetric transferability patterns and identifies key geographic regions (e.g., Prairie Gateway) where models fail, providing actionable guidance for future model design and data collection strategies.

Weaknesses:

1. The authors only conducted experiments with ViT and GNN-RNN structures. Some OOD techniques are also recommended to be implemented into this work to exam how these related works perform on the task of interests, e.g., [1], [2], and [3].

[1] Liu, W., Wang, X., Owens, J., & Li, Y. (2020). Energy-based out-of-distribution detection. Advances in neural information processing systems, 33, 21464-21475.

[2] Zhang, Y., Lu, J., Peng, B., Fang, Z., & Cheung, Y. M. (2024). Learning to shape in-distribution feature space for out-of-distribution detection. Advances in Neural Information Processing Systems, 37, 49384-49402.

[3] Li, T., Pang, G., Bai, X., Miao, W., & Zheng, J. (2024). Learning transferable negative prompts for out-of-distribution detection. In Proceedings of the IEEE/CVF Conference on Computer Vision and Pattern Recognition (pp. 17584-17594).

2. The dataset is limited to U.S. counties; cross-country or cross-crop experiments could enhance global relevance. While spatial OOD is well-explored, the temporal generalization (year-to-year transfer) could be analyzed more rigorously, e.g., assessing model drift under climate trend shifts or extreme weather conditions.

3. The paper reports average performance degradation but does not deeply analyze variance, uncertainty, or confidence calibration under OOD shifts.

4. Lack of qualitative results and failure cases. The authors are suggested to provide more qualitative results and the analysis of the underlying limitations based on more failure cases.

---

### Decision · Action_Editor_N7Ph · 2026-01-22

**Recommendation:** Accept with minor revision

**Additional Comments:**

I support the (split) majority decision of the reviewers to accept the paper. The submission makes a solid contribution for ML benchmarking and is supported by a comprehensive experimental evaluation.
However, the final revision should carefully address all reviewer comments. In particular, the authors are expected to improve the paper’s readability, add missing citations and references where appropriate, include proper table references, and implement the various editorial changes requested by Reviewer DhdB.

Final Decision: Accept with Minor Revisions.

**Audience:**

Yes

**Audience Explanation:**

The paper would be of interest to at least a subset of the TMLR audience, particularly researchers working on spatio-temporal prediction with ML models and especially on crop yield prediction.
It provides a thorough evaluation of state-of-the-art models and offers valuable insights by identifying key challenges related to OOD generalization.

**Claims And Evidence:**

Yes

**Claims Explanation:**

The claims regarding the systematic evaluation of out-of-distribution (OOD) generalization in crop yield prediction are well supported by accurate, convincing, and clearly presented evidence. The paper provides extensive experimental evaluations across multiple OOD settings.